# Temperature Dependence of Magnetization Dynamics in Co/IrMn and Co/FeMn Exchange Biased Structures

**Irina O. Dzhun [1],\*, Andrey V. Gerasimenko [2], Alexander A. Ezhov [3], Stanislav I. Bezzubov [4], Valeria V. Rodionova [5], Christina A. Gritsenko [5] and Nikolai G. Chechenin [1,3]**

1    Skobeltsyn Institute of Nuclear Physics, Lomonosov Moscow State University, 1/2 Leninskie Gory, 119991 Moscow, Russia; nchechenin@yandex.ru

2    Institute of Chemistry FEB RAS, 159 100-Letiya Vladivostoka Ave., 690022 Vladivostok, Russia; gerasimenko@ich.dvo.ru

3    Faculty of Physics, Lomonosov Moscow State University, 1/2 Leninskie Gory, 119991 Moscow, Russia; alexander-ezhov@yandex.ru

4    Kurnakov Institute of General and Inorganic Chemistry RAS, 31 Leninskii Pr., 119991 Moscow, Russia; bezzubov@igic.ras.ru

5    Research and Education Center "Smart Materials and Biomedical Applications", Immanuel Kant Baltic Federal University, 6 Gaidara Str., 236041 Kaliningrad, Russia; vvrodionova@kantiana.ru (V.V.R.); kbyrka@kantiana.ru (C.A.G.)

\*    Correspondence: dzhun@sinp.msu.ru

**Abstract:** Thin film ferromagnet/antiferromagnet (F/AF) exchange biased structures that are widely used in GMR spin valves are considered nowadays as promising systems for antiferromagnetic spintronic and spin-orbitronic devices. Here, the temperature dependences of magnetization dynamics in Co/IrMn and Co/FeMn F/AF structures are investigated using ferromagnetic resonance (FMR) in comparison to a free Co layer. A strong additional decrease in the resonance field was observed in Co/IrMn with a temperature decrease attributed to the rotatable anisotropy increase, which almost vanished at room temperature. In contrast to Co/IrMn, the contribution of the rotatable anisotropy in Co/FeMn is much weaker, even though it exists at RT, it is negative, and slightly varies with the temperature and resonance field shift in Co/FeMn. This is mainly due to unidirectional exchange anisotropy. FMR linewidth for the free Co layer increases with decreasing temperature and is accompanied with a slow relaxation process, while the additional contribution to FMR line broadening in Co/IrMn and Co/FeMn structures is correlated with variation in the exchange anisotropy. The observed results are discussed based on structural and surface morphology and magnetization reversal characterization using X-ray diffraction, atomic force microscopy, and vibrating sample magnetometry data.

**Keywords:** magnetization dynamics; ferromagnetic resonance; exchange bias; unidirectional anisotropy; rotatable anisotropy

## 1. Introduction

Ferromagnetic (F)/antiferromagnetic (AF) structures have been of great interest to researchers for several decades since the discovery of the exchange bias effect by Meiklejon and Bean in 1957. Along with the fundamental features of the occurrence of the exchange bias at the interface of F/AF layers, these structures are of interest due to their use in highly sensitive magnetic field sensors based on the effect of giant magnetoresistance. Also, in recent years, work has been carried out [1,2] to improve the sensitivity of magnetic sensors based on the effect of anisotropic magnetoresistance and the planar Hall effect [3,4], where F/AF structures also act as active elements. Active interest in the study of magnetization dynamics in such structures has now resumed due to the discovery of the spin Hall effect [5] and spin transfer [6,7] in some AFs. It is shown that FeMn and IrMn as AFs in

F/AF structures can act as spin current detectors in the inverse spin Hall effect [5,8] and as a source of spin current in the direct spin Hall effect [9]. In addition, the presence of an exchange bias in such structures leads to a spin-orbit torque-induced switching of the perpendicular magnetization of F under the presence of an electric current in the absence of an external magnetic field [10].

In FMR studies, magnetization dynamics is characterized by two main parameters: a resonance field that is inversely proportional to the spin precession frequency and FMR linewidth (LW) which characterizes the damping coefficient. Both are affected by many factors. Thus, for the free F layer the resonance field depends on the layer's saturation magnetization, surface anisotropy, and uniaxial magnetocrystalline anisotropy [11–13]. In the case of F/AF polycrystalline thin films, the structures resonance field is additionally affected by exchange bias due to the exchange interaction with AF spins and by rotatable anisotropy due to the presence of small unstable AF grains in which the AF magnetization rotates irreversibly as the FM magnetization is in rotation [14–17]. FMR linewidth in the F and F/AF structures (summarized in [18]) is more actively studied due to importance of the damping coefficient in spintronic devices. The main factors that determine FMR linewidth include the intrinsic Gilbert damping, two magnon scattering (TMS) on random magnetic fields created by a layer's surface roughness as well as the anisotropy axis direction spread [19–21], and the mosaic structure of the sample that assumes inhomogeneous variation of different parameters within the sample [22]. FMR linewidth can also be affected by the presence of impurities, dopants [23], presence of a spin pumping effect [24–26], and Eddy's current (which is negligibly small in thin F films). In F/AF structures, the FMR line is known to be much broader than in free F films [27,28]. It is commonly described in terms of the additional TMS on random fields that occurs due to local variation of the AF exchange bias field [29,30]. To separate the effects of this large number of factors from each other, the angle, temperature, and F [25] and AF [22,31] layers' thicknesses dependences of magnetization dynamics were studied.

Among these dependencies, the temperature factor is the least studied within the published literature. Several works have been published where F layers were not pinned with AF [24,28,32–34] and where either a slight change or absence in temperature variation of the magnetization dynamics were observed. The change in magnetization dynamics at a temperature decrease is attributed to the increase of the F layer's magnetization (for the FMR resonance field) and due to the relaxation process of impurities and dopants [28,33,34] in FMR linewidth. However, the possible changes in the surface and uniaxial anisotropies of the F layer in relation to the temperature were not considered.

The temperature dependence of magnetization dynamics in F/AF exchange biased structures is still not completely studied. It has been shown that the common features of such structures at a temperature decrease comprised a decrease in the FMR resonance field (called a left shift) and a huge FMR line broadening [35–39]. These observations were explained in terms of the slow relaxing process [40,41]; however, these works obtained different temperature dependencies of relaxation time.

Thus, a more systematic study of magnetization dynamics in F/AF exchange biased structures is required. As discussed above, the contribution of peculiarities in the F layer due to impurities and temperature dependence of uniaxial anisotropy and magnetization in relation to the temperature dependence of magnetization dynamics can be essential and should not be neglected. We assume that it should be included as an additional factor of F/AF magnetization dynamics that was not considered in the papers mentioned above. Rotatable and unidirectional anisotropies in the F/AF system, corresponding to presence of the AF layer, also substantially affect the FMR resonance field and linewidth. Additionally, the variation of these anisotropies in relation to temperature can be important in the temperature dependence of magnetization dynamics. These anisotropies, along with the low temperature relaxation process, are also dependent on F and AF layer microstructure. However, in most of the works mentioned above, no microstructure investigations were performed.

In this paper, we are trying to contribute to the knowledge of the influence of the factors discussed above on magnetization dynamics of F/AF systems. We report on systematic experimental study of the temperature dependence of FMR magnetization dynamics along with room temperature magnetostatic (VSM) characteristics in F/AF systems with the two most popular AF materials (IrMn and FeMn) compared to a free F layer, keeping the same thickness of the AF-layers and thickness of the Co as-F-layer, comparing the structural (XRD and AFM) features. Unexpected results are obtained and discussed in partitioning of the exchange coupling at the F/AF interface on exchange unidirectional and rotatable anisotropies and relative influence of these anisotropies on FMR damping and line broadening.

## 2. Materials and Methods

Experimental samples of Si/Ta 30 nm/Co 7 nm/Ir$_{45}$Mn$_{55}$ (Fe$_{50}$Mn$_{50}$) 15 nm/Ta 30 nm and Si/Ta 30 nm/Co 7 nm/Ta 30 nm were deposited on Si(100) substrate coated with native SiO$_2$ via DC magnetron sputtering in Ar at a pressure of $3 \times 10^{-3}$ Torr at ambient temperature in presence of 420 Oe magnetic field applied in plane of substrate. FMR measurements were performed at Brucker ELEXSYS e500 spectrometer. The sample was placed into cryostat and the temperature was changed by blowing it with liquid nitrogen vapor to cool it down from room temperature to 115 K. The cryostat with the sample was set in the resonator supplied with the fixed pumping frequency $f = \omega/2\pi = 9.4$ GHz. The resonator was placed between the electromagnet poles, provided the slowly varied magnetic field aligned along the sample surface plane, and was scanned up to 1500 Oe to reach the resonance condition. X-ray diffraction (XRD) investigations were made using RIGAKU SmartLab diffractometer (Japan) (9 kW rotating anode), CuKα-radiation, parallel beam (CBO optics), 2θ mode, silicon zero-background specimen holder, and HyPix–3000 detector (1D measurement mode). The diffraction pattern was recorded within the 2θ range of 10°–80°, at the 2θ step of 0.01°. The XRD peaks were identified using the ICDD PDF-2 and ICSD databases. Hysteresis loops were obtained at Lake Shore VSM magnetometer at ambient temperature. To investigate roughness of the layers' topography and magnitude (error), signal images were obtained by atomic force microscopy (AFM) using a scanning probe microscope, NTEGRA Prima (NT-MDT, Moscow, Russia), operated in a semi-contact mode with a 5–10 nm peak-to-peak amplitude of "free air" probe oscillations. Silicon cantilevers NSG01 "Golden" series cantilevers for semi-contact mode (NT-MDT, Moscow, Russia) and PointProbe® Plus AFM Tips PPP-NCH-20 (Nanosensors, Neuchatel, Germany) were used. Image processing was performed using the Image Analysis software (NT-MDT, Moscow, Russia). The magnitude (error) signal images are shown because they are not affected by slope on scales comparable to the side of the image and do not provide "contrasts" within the details of the image. The grain size distributions were manually obtained from AFM images using ImageJ 1.50b software.

## 3. Results

### 3.1. Structure Properties

XRD spectra of the samples at $\theta/2\theta$ geometry and at grazing incidence (GI) or at a grazing angle of 1° are shown in Figure 1. To reveal the XRD features of the substrate with the buffer layer, the Si/Ta 30 nm sample was also prepared and measured.

One can see clear diffraction peaks corresponding to Co (111), FeMn (411), and IrMn (101). No other texture peaks corresponding for these materials were observed in the whole investigated angle range. Grain size in the F and AF layers is an important factor of influence on the micromagnetic properties [42] and was evaluated using Sherrer's equation, which was applied to the XRD peaks and was 33 nm for Co, 19 nm for FeMn, and 13 nm for IrMn. The grain size in such polycrystalline structures is commonly considered to be in the order of the film's thickness, such as for the AF layers in our case. Large crystallite size in the Co layer at low film thickness is surprising but can be explained by the heredity of grain size of the thick underlying Ta buffer layer. The lower intensity of the FeMn texture

peak in contrast to the IrMn indicates that this layer is not well textured. The absence of the Co (111) peak along with any other Co peaks in the GI spectrum can be also due to the absence of prevailing texture in the film. With regard to the Ta buffer layer, the same diffraction peaks were also observed in [43] when Ta was sputtered onto an SiO$_2$ substrate and was attributed to a Ta (100) textured film. The different intensities of Ta diffraction peaks, as demonstrated in Figure 1c, at the same layer thicknesses, can be attributed to different textures of FeMn and IrMn underlying layers.

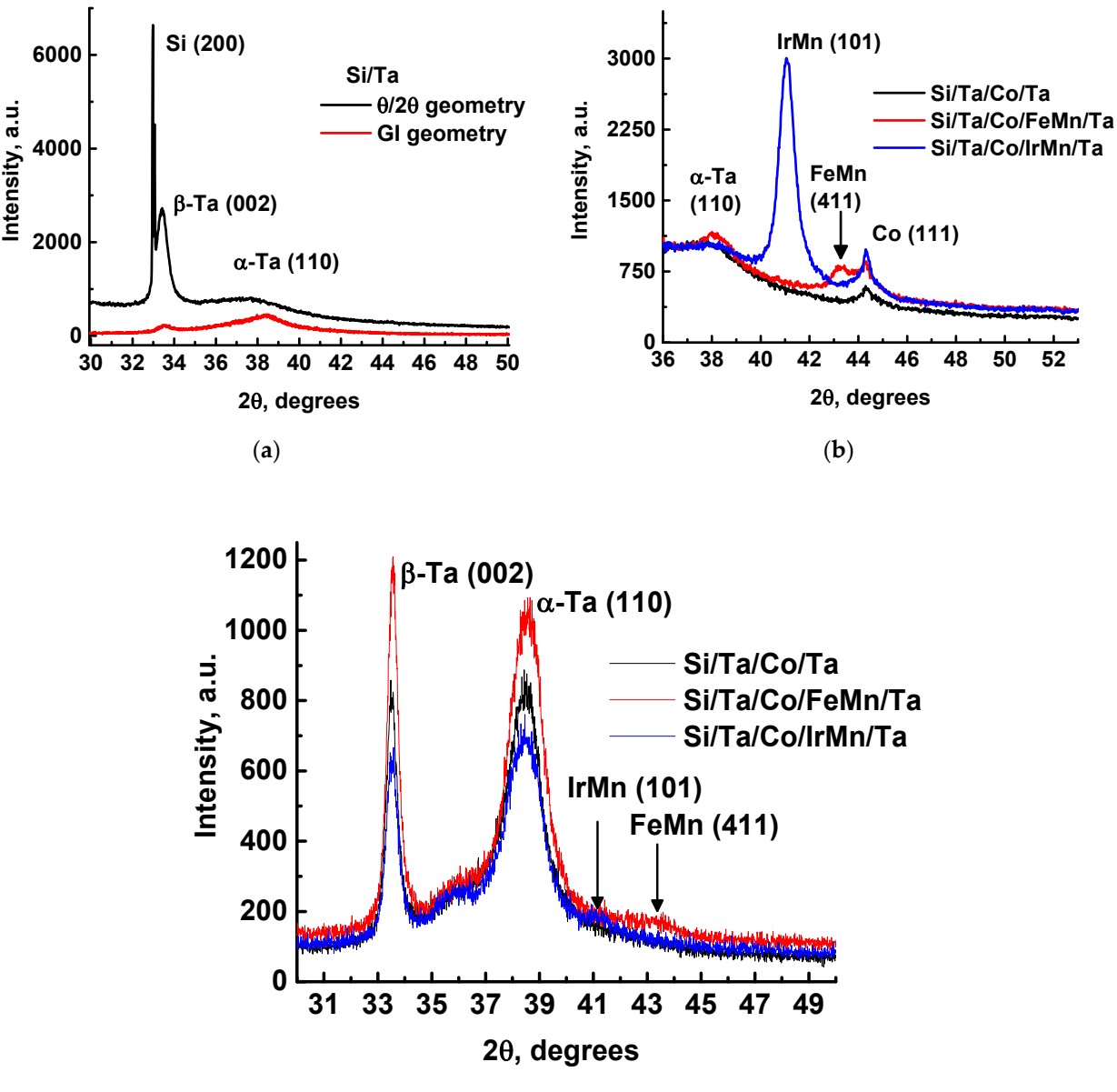

**Figure 1.** XRD spectra at θ/2θ (black lines) and GI (red line) geometry for substrate with buffer Ta layer (**a**), XRD spectra at θ/2θ (**b**), and GI (**c**) geometry for Co/FeMn, Co/IrMn, and free Co layer.

### 3.2. Roughness and Graininess

For this investigation, samples of Si/Ta, Si/Ta/Co, Si/Ta/Co/FeMn, and Si/Ta/Co/IrMn without capping of the Ta layer were deposited. AFM topography images and corresponding magnitude (error) signal images of each layer are shown in Figure 2.

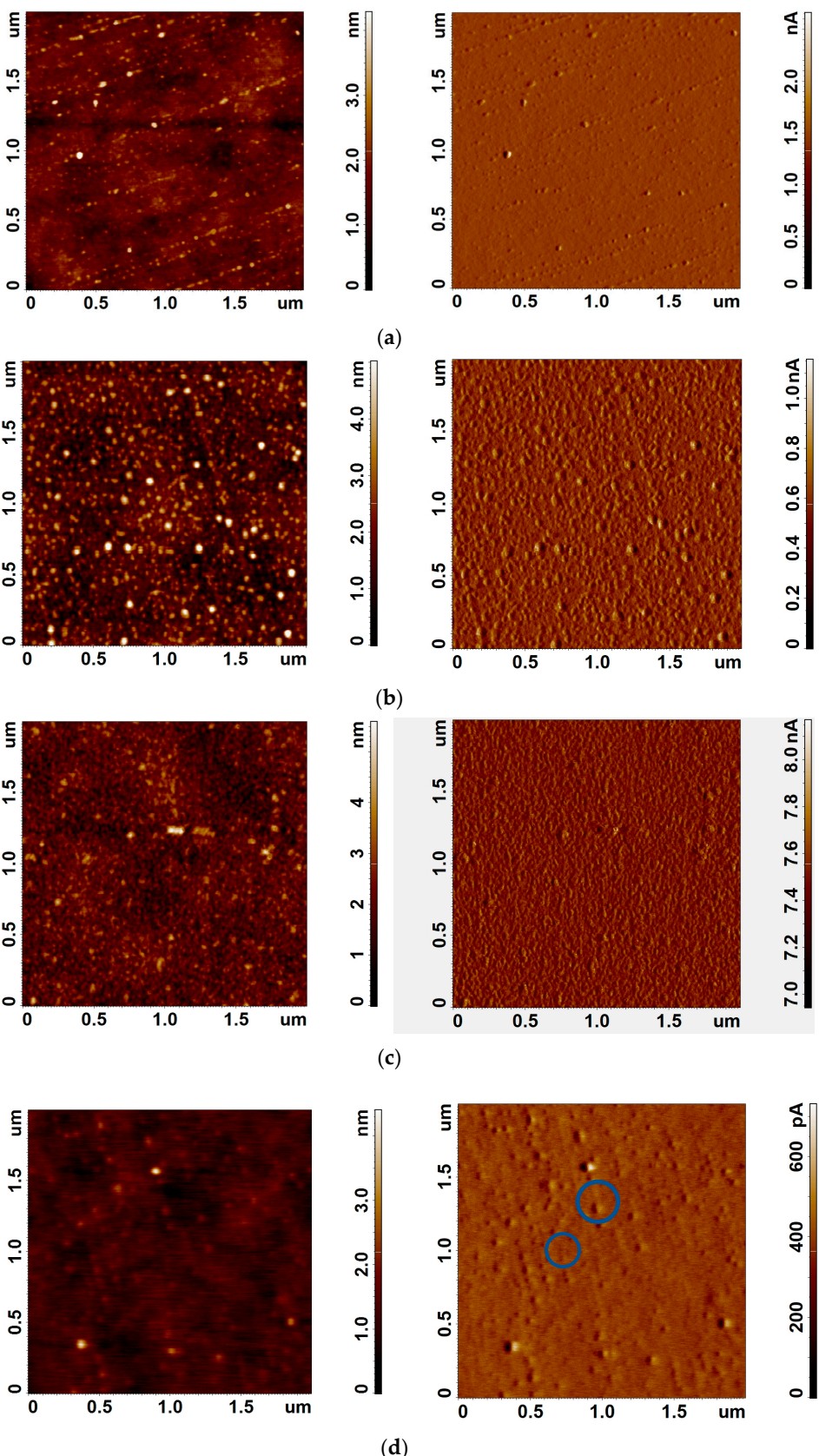

**Figure 2.** Topography (**left**) and magnitude (**right**) AFM images of Si/Ta (**a**), Si/Ta/Co (**b**), Si/Ta/Co/IrMn (**c**), and Si/Ta/Co/FeMn (**d**) samples. Examples of larger and smaller grains in FeMn layer are circled.

The AFM results in Figure 2 qualitatively confirm a larger grain size in the Co layer compared with the IrMn layer. The FeMn film's surface also demonstrates the presence of grains larger than in IrMn. The grain size distributions for Co, FeMn, and IrMn layers along with the mean grain size obtained from lognormal fit are presented in Figure 3. The quantitative difference between grain sizes obtained from XRD and AFM data can be due to low depth of frontiers between grains that inhibits obtaining proper grain size values. Low density of large (clearly visible on AFM images) grains in the FeMn layer along with their chaotic nonuniform distribution can be a consequence of poor film texture confirmed by XRD data and the presence of broad grain size distribution.

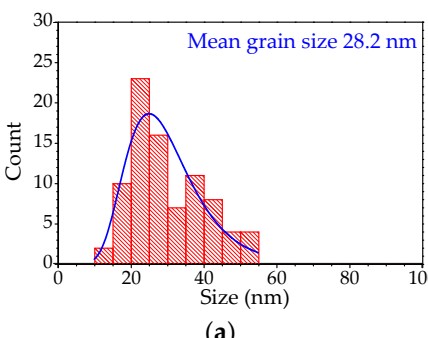 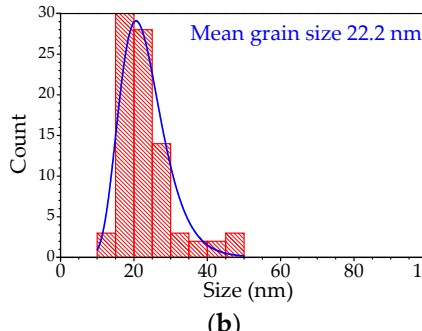 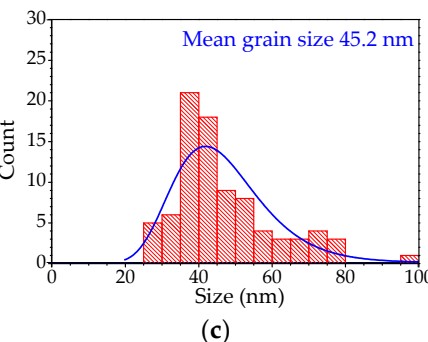

**Figure 3.** Statistic grain size distribution and mean grain size for Si/Ta/Co (**a**), Si/Ta/Co/IrMn (**b**), and Si/Ta/Co/FeMn (**c**) samples. Blue solid lines are lognormal fits.

The root mean square and average roughness of each layer measured in the 4 $\mu m^2$ sample's area is given in Table 1. Both RMS and average roughness have the highest value for Co film. The decrease of the FeMn layer roughness can also be associated with poor film texture [44].

**Table 1.** Root mean square and average roughness of the samples.

| Sample | RMS, nm | Ra, nm |
|---|---|---|
| Si/Ta | 0.47 | 0.36 |
| Si/Ta/Co | 0.71 | 0.53 |
| Si/Ta/Co/IrMn | 0.54 | 0.41 |
| Si/Ta/Co/FeMn | 0.27 | 0.20 |

### 3.3. Magnetostatic Measurements

The hysteresis loops for free and pinned Co layers obtained by in-plane measurements along the easy axis (EA) and hard axis (HA) directions are shown in Figure 4.

One can see that the coercivity values along and perpendicular to the EA directions for free Co layers are both equal to 28 Oe, while the ratio of residual magnetization $Mr = M(0)$ to the saturation magnetization $Ms$ is different which means that the presence of uniaxial magnetocrystalline anisotropy can be estimated from the hysteresis loop, as shown in Figure 4a. Coercivity of the Co/IrMn structure along the EA is very close to that of free Co (24 and 28 Oe, respectively) and significantly lower (7 Oe) that that in the hard axis (HA) direction. The Co/FeMn sample is characterized by a coercivity increase up to 50 Oe along the EA. The hysteresis loop along the HA is not presented because the magnetometer's field is not sufficient to saturate the F layer in this direction. The values of exchange bias, coercivity, and uniaxial magnetocrystalline field $H_K$ for the samples are summarized in Table 2, together with the results obtained from FMR.

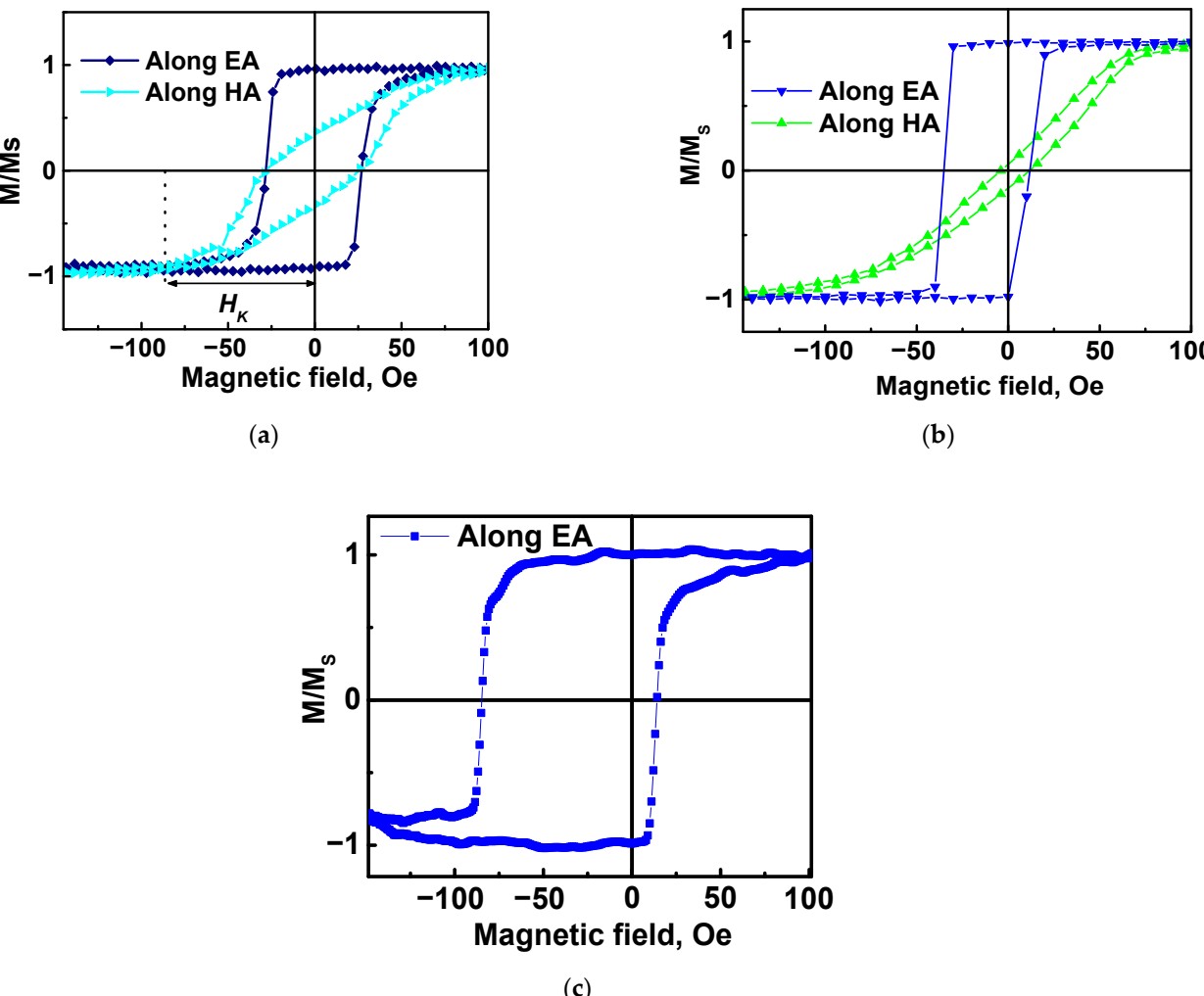

**Figure 4.** Hysteresis loops for free (**a**) and pinned with IrMn (**b**) and FeMn (**c**) Co layers.

**Table 2.** The room temperature values of uniaxial anisotropy and exchange bias fields obtained by different techniques and coercivity along EA and HA.

| Sample | $H_{EB}$ FMR/VSM, Oe | $H_K$ FMR/VSM, Oe | Hc EA/HA, Oe |
|---|---|---|---|
| Free Co | - | 82/86 | 28/28 |
| Co/IrMn | 12/14 | 73/88 | 24/7 |
| Co/FeMn | 0/35 | 60/- | 50/- |

### 3.4. Ferromagnetic Resonance

Typical FMR spectra for the Co/IrMn structure at room temperature and 150 K obtained along the HA direction are shown in Figure 5. One can see, from the FMR spectra (Figure 5), an increase of the peak-to-peak linewidth ΔW along with a left shift in the resonance field at the temperature decrease that is typically observed for exchange biased structures.

Temperature dependencies of FMR fields for free Co, Co/FeMn, and Co/IrMn obtained by in-plane measurements at different EA orientations are shown in Figure 6. For the free F layer, the resonance field along the EA slightly decreases from 494 Oe at RT to 479 Oe at 115 K. At the same time along the HA direction, an unusual slight increase of the FMR resonance field is observed. For the Co/IrMn sample, the FMR resonance field shows a rapid monotonic decrease for all sample orientations. At the same time, for the Co/FeMn sample, the behavior of *Hr* temperature dependence is quite different for different sample

orientations as follows: at $\alpha = 0°$, 90° and 270° a slight decrease of $Hr$ is observed, while at $\alpha = 180°$ the $Hr$ rapidly increases, especially in the 150–115 K temperature range.

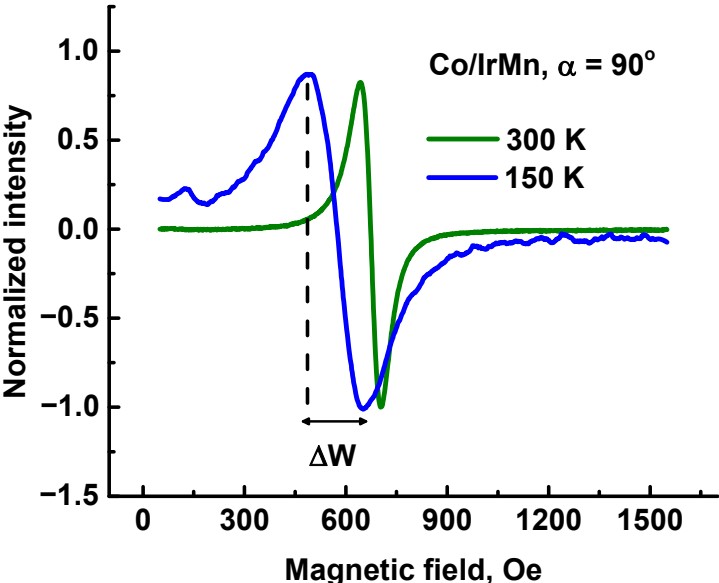

**Figure 5.** FMR spectra of Si/Ta/Co/IrMn/Ta sample obtained at RT and 150 K by in plane measurements along HA direction.

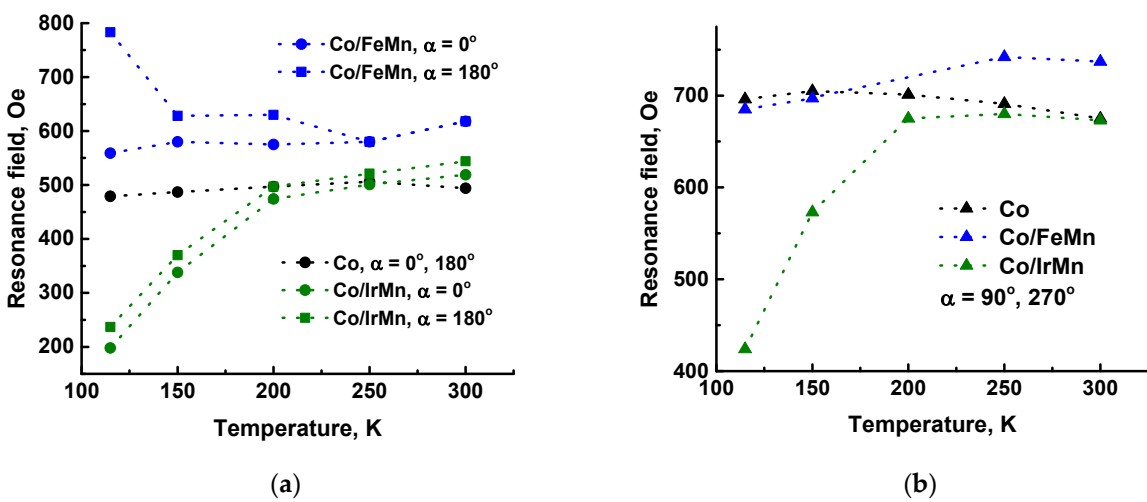

      (**a**)      (**b**)

**Figure 6.** Temperature dependencies of FMR resonance field for Co, Co/IrMn, and Co/FeMn structures obtained along (**a**) and perpendicular (**b**) to EA directions.

The in-plane angular distribution of the FMR resonance field in the simplest way can be expressed for exchange biased structures (i.e., with account of exchange interaction between F and AF layers) as

$$Hr = H_{iso} - H_{EB}\cos\alpha - H_K\cos2\alpha, \tag{1}$$

where $H_{EB}$—exchange bias, $H_K$—uniaxial anisotropy fields and

$$H_{iso} = (\omega/\gamma)^2/4\pi M_{eff} - H_{RA}, \tag{2}$$

is the isotropic angular independent term, in that $H_{RA}$ is rotatable anisotropy field aligned along the applied field. From (1) and (2) one can see that the exchange coupling effect of

the AF layer on the F-layer at the F/AF interface is double-sided. On one hand, exchange coupling gives a well-known unidirectional exchange biasing with the field $H_{EB}$, and, from the other side, it manifests itself as a rotatable anisotropy field $H_{RA}$. Since there is no exchange coupling for the free F layer, i.e., $H_{EB} = H_{RA} = 0$, we can estimate the $H_{RA}$ field from (2), assuming the rest of the characteristics are the same for biased and unbiased systems, as the difference of $H_{RA} = H_{iso}(\text{F/AF}) - H_{iso}(\text{free F})$ [14].

The values of exchange bias and uniaxial anisotropy extracted from FMR data at ambient temperature are presented in Table 2 and compared to that obtained from VSM. One can see that FMR gives close data for "stationary" characteristics which can be obtained from VSM for the free Co layer and Co/IrMn structures. For the Co/FeMn sample, the results are surprisingly different. The reasons for such results are not clear and can probably be related to the difference between magnetostatic and magnetodynamic methods.

Temperature dependencies of FMR linewidths are shown in Figure 7. The LW increases with decreasing temperature for all samples. For IrMn-biasing, the LW at RT is close to that of the unbiased Co sample but increases more rapidly. In the FeMn-biased sample, the LW is larger than that in the free Co, even at RT (107 Oe vs. 57 Oe, respectively). Also, one can note from Figure 6 that LWs have a weak angular dependence, excluding the Co/FeMn sample shows a significant increase at a temperature below 150 K for the reversed ($\alpha = 180°$) applied field, i.e., for the direction with the largest exchange biasing.

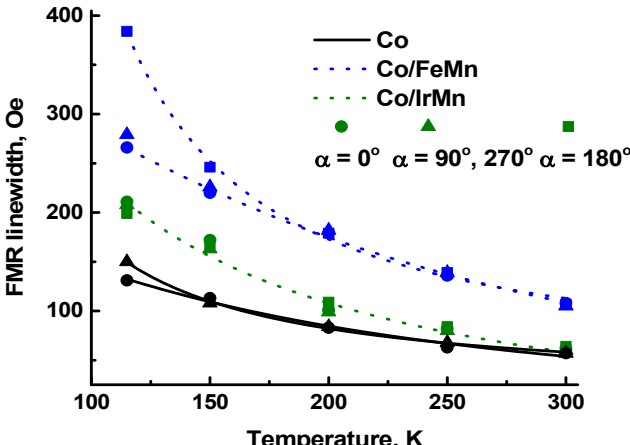

**Figure 7.** Temperature dependencies of FMR linewidths for Co/IrMn, Co/FeMn, and free Co layer obtained at different EA orientations. Black solid lines fit with Equation (2). Dotted lines are eye guides.

## 4. Discussion

Effective magnetization of the free Co layer extracted from Equation (1) with $\gamma = 1.845$ [20] vs. temperature is presented in Figure 8a. On the same graph, temperature dependence of saturation magnetization of bulk Co using the Bloch's relation

$$M_S(T) = M_S(0) \times (1 - (T/T_C)^{3/2}), \tag{3}$$

where $Ms(0)$ is spontaneous magnetization at absolute zero and $T_C$ is Curie temperature is also plotted.

The effective magnetization is below the bulk $M_S$ by about 1.5 kGs, which can be related to the presence of the out-of-plain surface anisotropy field $Hn = 2Kn/Ms$ as in (4).

$$4\pi Meff = 4\pi Ms - 2Kn/Ms, \tag{4}$$

where the surface anisotropy $Hn$ pushes magnetization vector $M$ out of the film's plane as suggested in [19,45].

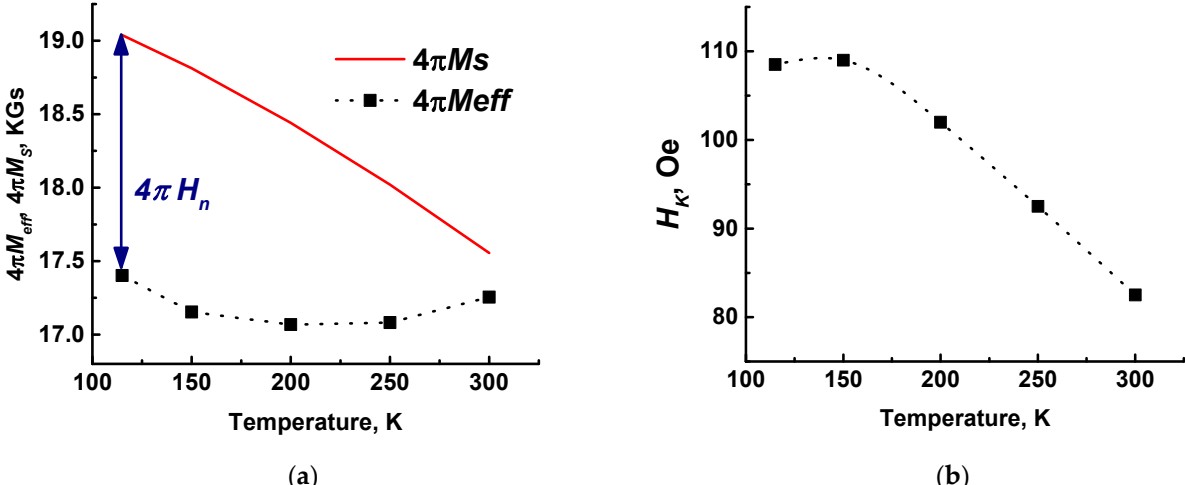

**Figure 8.** Temperature dependencies of effective magnetization (**a**) and uniaxial anisotropy (**b**) of free Co layer.

An alternative explanation of lowering the effective magnetization is an angular spread of magnetic moment due to an angular spread of uniaxial anisotropy in nanocrystalline microstructure of Fe-layers [46]. A non-complete averaging of magnetocrystalline anisotropy during the F-layer deposition causes residual spread of uniaxial anisotropy around the EA, leading to a stray field and local oscillation of magnetization [47,48], which is observed as a ripple structure in a defocused film image in the Lorentz transmission electron microscopy (LTEM) in the Fresnel mode [49–51]. We have demonstrated that the larger the GS is the larger the local angular dispersion of magnetization is [52].

It was shown that the difference between $M_{eff}$ and $M_S$ can be essential in thin Co [24] and Py [25] films. It can be also one of the reasons of decrease of $H_K$ at the temperature increases (Figure 8b) [45]. Thus, the temperature dependencies of free Co layers in easy and hard axis directions are mainly caused by competing surface and uniaxial anisotropies. That can also be the reason for the slight increase of the HA resonance field at a temperature decrease from 300 to 150 K. Such an increase was also observed in [45] in the case of thin capping layers and in [24] at a Co thickness above 7 nm and is attributed to $4\pi M_{eff}$ dispersion.

The temperature dependencies of the isotropic FMR field term $H_{iso}$ for Co/IrMn, Co/FeMn, and free Co layer (a) and rotatable anisotropy fields (b) are plotted in Figure 9.

The isotropic resonance field term in the Co/IrMn sample coincides with that of a free Co sample at ambient temperature, as shown in Figure 9a, indicating a negligible contribution of rotatable anisotropy; however, it arrives and grows with decreasing temperature, as shown in Figure 9b, by moving $Hr$ and $H_{iso}$ to the left side in accordance with Equations (1) and (2). The enhanced contribution of the $H_{RA}$ at a lower temperature can be interpreted as an antiferromagnetic analog to the superparamagnetic effect in ferromagnetic media. The antiferromagnetic layer can be assumed to be composed from grains of different sizes, roughly of relatively large and relatively small sizes. The magnetic moment of large stable AF grains at the F/AF interface is stable, and can withstand the change in orientation of the applied external magnetic field; thus, supplying the unidirectional exchange anisotropy bias, represented by field $H_{EB}$ in Equation (1). The smaller AF grains have a weaker stability to withstand the thermal effect from one side and a magnetizing influence of the applied field from the other side (thermally disordered grains). The lower the temperature the larger fraction of grains, which can withstand the thermally activated magnetic disordering, keeping their magnetic moment at the interface in contact with the F-layer along the direction of the applied field, composing the rotatable magnetic field, $H_{RA}$. One can conclude from Figure 9b that the fraction of such grains is relatively large in IrMn at a low temperature, but vanishes approaching the RT.

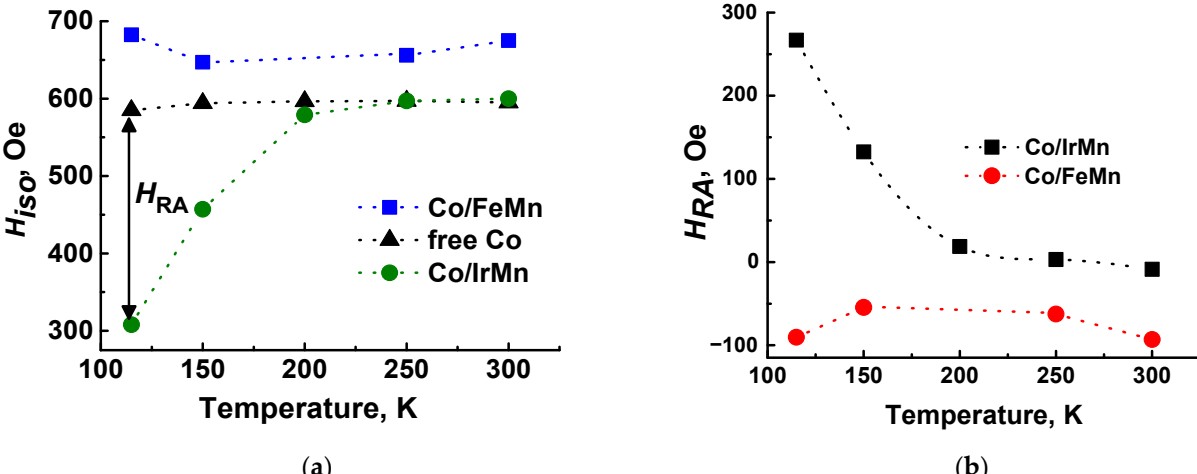

**Figure 9.** Temperature dependencies of $H_{iso}$ for Co/FeMn, Co/IrMn, and free Co layer (**a**) and rotatable anisotropy field for Co/FeMn and Co/IrMn (**b**).

In contrast, in Co/FeMn, the AF has an effect on the F layer, but enhances resonance field value, as shown in Figure 7a, indicating the negative contribution of rotatable anisotropy, as shown in Figure 9b. The temperature dependence of $H_{iso}$ and rotatable anisotropy in Co/FeMn are totally different from that in Co/IrMn. A shift of $H_{iso}$ to a higher value, which can be attributed to the negative rotatable anisotropy, is large even at RT, as shown in Figure 9. The temperature dependence is not so sharp as in Co/IrMn and is not monotonous.

Negative rotatable anisotropy was observed earlier in structures with IrMn and FeMn [53–55]. As discussed in these works and schematically summarized in [56], negative rotational anisotropy appears when small unstable AF grains are antiferromagnetically coupled (AFC) to F magnetization. As in [53], we can attribute negative $H_{RA}$ to a presence of a fraction of unstable AFC interfacial grains with uncompensated spins in the FeMn layer. As discussed in [53], the appearance of AFC grains depends on the texture and interface distance of the layers. In our case, that can probably be also due to the poor texture of the AF layer along with the high interface roughness (high underlying Co layer roughness) in this sample or in the assumption of coexistence of small and large grains in the FeMn layer, as follows from our XRD and AFM results.

The non-monotonic behavior of the temperature dependence of $H_{RA}$ in Co/FeMn sample (Figure 9b) can be qualitatively explained with a presence of two fractions of grains as follows: At room temperature the smaller grains are thermally disordered and act as paramagnetic dopants. The larger grains are magnetically ordered but still unstable that results in the presence of rotatable anisotropy at the absence of exchange bias at RT FMR data and in coercivity increase. At the temperature decrease, these larger grains become stable and begin to contribute to exchange bias instead of $H_{RA}$; this leads to $H_{RA}$ decrease and appearance of nonzero $H_{EB}$ [57]. As the temperature further decreases the smaller thermally disordered AF grains become ordered but unstable and begin to contribute to $H_{RA}$ so that the absolute value increases again, as seen in Figure 9b. That can also be a reason for non-typical exchange bias manifestation in the Co/FeMn sample. Thus, from Equation (1), the exchange bias can be obtained as $2 \times H_{EB} = Hr(\alpha = 180°) - Hr(\alpha = 0°)$. The increase of exchange bias at the temperature decrease is observed as a more rapid decrease of $Hr\ \alpha = 0°$ and a slight decrease of $Hr(\alpha = 180°)$ (as in our results for Co/IrMn or more clearly seen, for example, in [58]). In case of Co/FeMn (Figure 6a), we observe a slight decrease in $Hr(\alpha = 0°)$ along with a sharp increase in $Hr(\alpha = 180°)$; this can be caused by unstable AFC grains that become stable but still AF coupled.

In contrast to Co/FeMn, the monotonic increase of $H_{RA}$ in Co/IrMn can be attributed to the lack of unstable AF grains at room temperature due to uniform grain size distribution

and higher values of the AF anisotropy constant and AF stiffness of the IrMn layer. The higher value of $H_{RA}$ in Co/IrMn compared to Co/FeMn at low temperatures can be assigned to a large fraction of small thermally disordered grains at RT in the IrMn layer.

Temperature dependence of FMR linewidth of the free Co layer in Figure 7 is in good agreement with the slow relaxing process due to spin-lattice relaxation via impurity interactions with conduction electrons described in [33]. The solid line in Figure 7 is a fit with the following equation:

$$\Delta W = B + \frac{A}{T} \frac{\omega\tau}{1 + (\omega\tau)^2} \tag{5}$$

where relaxation time $\tau = 1/cT^2$. The obtained values of parameters A ($2.1 \times 10^4$) and $c/2\pi$ ($0.6 \times 10^{-3}$ GHz/K$^2$) are close to that in [33].

The broadening of FMR linewidth (Figure 7) in the Co/FeMn sample in contrast to Co/IrMn and the free Co samples at room temperature can be explained by poor a FeMn layer texture accompanied with inhomogeneous distribution of large grains that lead to the increase in the mosaic term and also resulted in fluctuations in exchange coupling that can also broaden $\Delta W_{F/AF}$ via the TMS mechanism [30]. Thus, FMR line broadening along with a large rotatable anisotropy field and absence of exchange bias at RT FMR measurements is mainly caused by TMS and mosaic mechanisms [22]. Both mechanisms are anisotropic and could result in FMR line anisotropy (Figure 7).

From the discussion about the temperature dependence of $H_{RA}$ in the Co/IrMn sample, we concluded that it is attributed to a presence of a large amount of disordered small grains at RT in the IrMn layer. In this assumption, more rapid FMR line broadening at the temperature decrease in Co/IrMn, in contrast to free Co layer, seems to be interpreted in terms of the slow relaxing process with $\tau \sim T^{-2}$ for the presence of very small AF grains that act as paramagnetic-like dopants [40]. As described in [40], the FMR LW, in terms of this process, can be written in the same form as in Equation (5). On the other hand, the presence of the AF layer leads to an additional relaxation process with the Neel relaxation time $\tau \sim \exp(\Delta E / k_B T)$ [36] where $\Delta E$ is $K_{AF} \times V$; $K_{AF}$ is AF anisotropy constant ($3 \times 10^6$ erg/cm$^3$ for FeMn [41] and $5.25 \times 10^6$ erg/cm$^3$ for IrMn [57]); and V is the grain volume, which can be roughly evaluated as $l^3$ where $l$ is the grain size from XRD data. In our case, $\Delta E_{FeMn}/\Delta E_{IrMn} = 1.78$, which results in a higher $\tau$ for the Co/FeMn sample and, consequently, possibly greater line broadening, even at RT.

In our results, FMR linewidth temperature dependence for bilayer structures $\Delta W_{F/AF}(T)$ and linewidth difference $\Delta W_{F/AF} - \Delta W_F$ are not described in terms of a slow relaxation process neither with $\tau \sim T^{-2}$ nor with $\tau \sim \exp(\Delta E / k_B T)$. Moreover, in these relaxation mechanisms, relaxation time $\tau$ is derived by both a left shift of the FMR resonance field $S$ and FMR line broadening $\delta\Delta W$ as $\tau = -2S/\delta\Delta W$. In our case, it gives anisotropic relaxation times for different sample orientations and does not describe the right shift of $H_r$ at $\alpha = 180°$ for Co/FeMn at the temperature decrease. It seems to be a presence of an additional mechanism of $\Delta W_{F/AF}$ broadening that is associated with the increase of exchange bias at the temperature decrease. In Figure 10, the temperature dependences of line broadening difference $\delta(\Delta W/\Delta W_{RT}) = \Delta W/\Delta W_{RT}(F/AF) - \Delta W/\Delta W_{RT}(F)$ and normalizes exchange bias are presented. Comparing the tendencies of the curves in Figure 10, one can easily see a similarity between the temperature dependencies of the additional line broadening and exchange bias, especially at temperatures below 250 K. It has been shown in [27] that the exchange anisotropy directly affects FMR line broadening. Our results confirm this conclusion and, furthermore, we can state that its variation with the temperature should be accounted for in the temperature dependent damping mechanism.

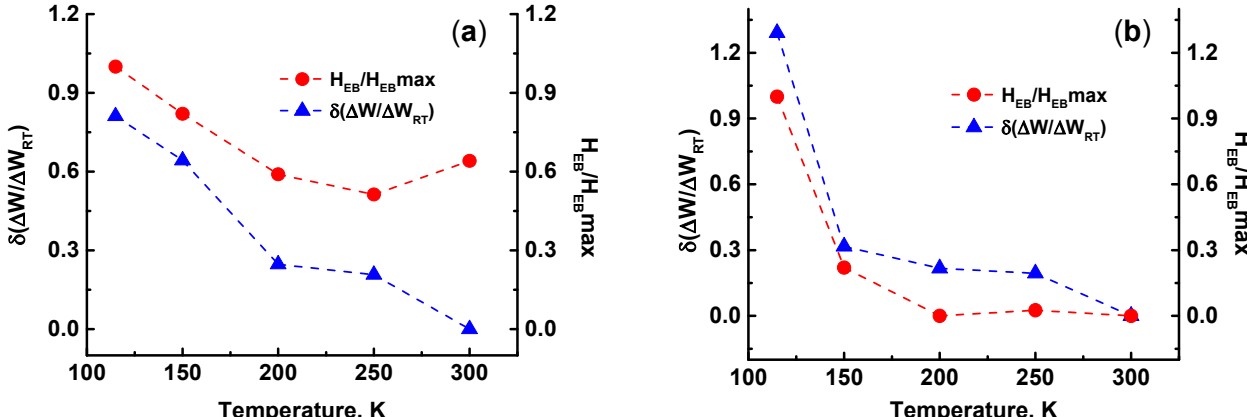

**Figure 10.** Temperature dependences of line broadening difference $\delta(\Delta W/\Delta W_{RT})$ (triangles) and normalized exchange bias (circles) for Co/IrMn (**a**) and Co/FeMn (**b**). Here, $H_{EB}$max = 19.5 Oe and 112 Oe for Co/IrMn and Co/FeMn, respectively.

However, the mechanism of exchange F/AF coupling on the LW broadening is not quite clear at this moment. It is important to note that LW enhancement is controlled by exchange bias to a larger extent than by rotatable exchange coupling as follows: at low temperature a broader LW correlates with larger $H_{EB}$ in Co/FeMn as shown above, and not with larger $H_{RA}$ as compared in Figures 9b and 10.

The anisotropy of linewidth broadening (Figure 11) with the temperature decrease for Co may be a consequence of two-magnon scattering on structural defects and uniaxial anisotropy dispersion of the F layer as discussed in [19,45]. On the other hand, it may be associated with the increase in uniaxial anisotropy at the temperature decreases (Figure 9b), but uniaxial anisotropy does not change from 150 to 115K while the anisotropy of FMR linewidth appears. Uniaxial anisotropy is related to a certain crystallographic direction (for example (111) for permalloy). From the XRD data, namely from disappearance of the Co peak in the GI geometry, we can assume that the texture of the F layer is practically absent in our sample, which leads to a high value in the $H_K$ dispersion.

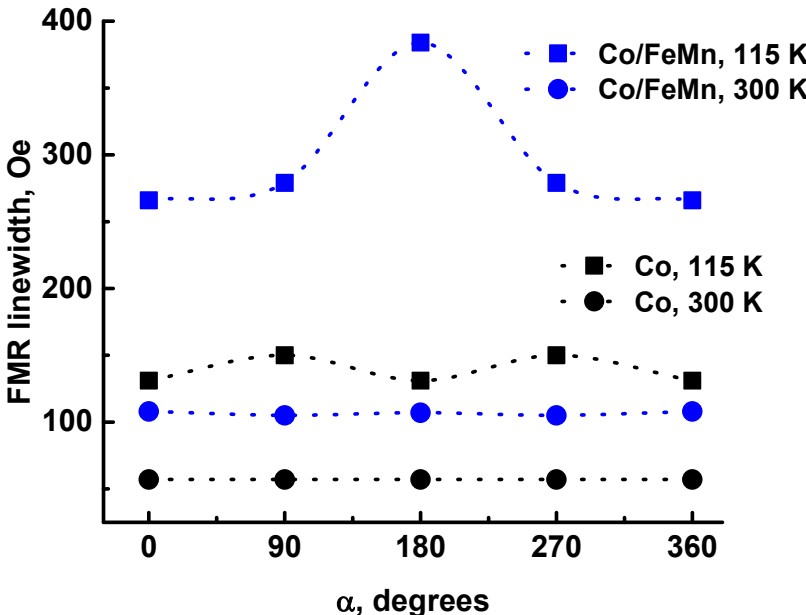

**Figure 11.** Angular distribution of FMR linewidth for Co and Co/FeMn samples at RT and 115 K.

In Co/FeMn the additional FMR LW broadening appears due to AF axis dispersion that is caused by a weak FeMn layer texture and inhomogeneous large grains distribution in contrast to Co/IrMn sample, where, along with high texture and uniform surface FMR LW broadening is close to being isotropic. The largest anisotropy of FMR LW in Co/FeMn at 115 K is also associated with a high value of exchange bias and confirms its contribution to the damping mechanism.

## 5. Conclusions

In the paper, we have demonstrated that FMR reveals two features of exchange coupling between AF and F layers at the interface of the F/AF bilayer system—a unidirectional anisotropy (normally called exchange bias, EB) and rotatable anisotropy (RA). In the pinned with IrMn Co layer, RA decreases the FMR resonance field when the temperature decreases, while the EB has a much smaller temperature effect on the FMR resonance field. In the Co/FeMn sample, RA increases the resonance field, even at room temperature. At the temperature decrease, the resonance field is affected by both RA and EB.

FMR LW is essentially isotropic at RT but shows anisotropy when lowering the temperature. LW broadens when the temperature decreases in both exchange biased and unbiased systems and this broadening increases in exchange biased systems. A larger LW correlates with larger $H_{EB}$ at the temperature decrease for both Co/FeMn and Co/IrMn. Rotatable anisotropy, as a form of exchange coupling, has a smaller effect on LW compared with unidirectional anisotropy.

We assume that most of the reported effects stem from the difference in microstructure, i.e., grain size distribution and texture of the samples.

The results reported here demonstrate a powerful ability of FMR in the study of dynamic effects in ferromagnetic media, especially in the exchange coupled F/AF system, when combined with traditionally magnetostatic tools, such as VSM, and microstructural instrumentation. FMR reveals a branching of exchange coupling to the rotatable and static unidirectional exchange interaction at the F/AF interface and allows a deep study of this exciting phenomenon.

**Author Contributions:** Conceptualization, I.O.D. and N.G.C.; methodology, N.G.C.; validation, I.O.D. and S.I.B.; formal analysis, I.O.D.; investigation, I.O.D., A.V.G., A.A.E., V.V.R. and C.A.G.; data curation, I.O.D., A.V.G., A.A.E., S.I.B., V.V.R. and C.A.G.; writing—original draft preparation, I.O.D., A.V.G., S.I.B. and A.A.E.; writing—review and editing, N.G.C.; visualization, I.O.D. and A.A.E.; supervision, N.G.C.; project administration, N.G.C. All authors have read and agreed to the published version of the manuscript.

**Funding:** This research received no external funding.

**Institutional Review Board Statement:** Not applicable.

**Informed Consent Statement:** Not applicable.

**Data Availability Statement:** The data used in this research are available from the corresponding author upon reasonable request.

**Acknowledgments:** This work was carried out within the frame of priority research direction #8 of the SINP MSU, project #122081700088-9. The additional hysteresis loop measurements were performed at the Laboratory of Electrodynamics and Microwave Electronics, Kirensky Institute of Physics SB RAS, Krasnoyarsk, Russia. The authors wish to acknowledge Stanislav Zabotnov, Dmitrii Shuleiko, and Nikolay Galkin for fruitful discussions.

**Conflicts of Interest:** The authors declare no conflict of interest.

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
