# Peer review of "Temperature Dependence of Magnetization Dynamics in Co/IrMn and Co/FeMn Exchange Biased Structures"

_magnetochemistry, doi:10.3390/magnetochemistry9100218_

Round 1
Reviewer 1 Report
The manuscript “magnetochemistry-2545292” deals with the thin film ferromagnet/antiferromagnet Co/IrMn and Co/FeMn systems. All the results which are discussed in this work are based on the structural, surface morphology, and magnetization reversal characterization using X-ray diffraction, atomic force microscopy, and vibrating sample magnetometry data. I have a few questions before accepting the manuscript which are mentioned below:
1. What was the main motivation behind this work is not clear. I need to understand what is the novelty of the work. What is an important outcome of this work which is different from the earlier studies?
2. Can authors provide insight into how the results change with varying thicknesses of FM and AFM layers?
3. Fig. 5a shows the opposite behavior for Ir and Fe case at a temperature less than 200K for the resonance field. What is the reason behind this? Both, these are AFM layers.
The English is good.
Author Response
We thank the reviewer for high evaluation of the importance of our manuscript and helpful questions.
Point 1: What was the main motivation behind this work is not clear. I need to understand what is the novelty of the work. What is an important outcome of this work which is different from the earlier studies?
Response 1: Thanks for your question. Nowadays there is insufficient array of experimental data and publications on the temperature dependence of magnetization dynamics in F/AF exchange biased structures. Publications, where the FMR damping mechanisms are studied, are lacking the temperature effects, and vice versa, where the temperature effects were investigated, the mechanisms are considered which are not applicable to our systems. Besides in the most of these studies no microstructure investigations were performed.
In our work we report on a detailed study on low temperature magnetization dynamics in F/AF structures compared with that for free F layer, along with their room temperature magnetostatic and microstructure characteristics. In this paper we report on the qualitative relation between samples microstructure and their temperature dependence of FMR resonance field and linewidth. From consideration of free F layer we show that its peculiarities such as increase of uniaxial anisotropy value at the temperature decrease, are essential for the temperature dependencies of FMR linewidth and resonance field. In F/AF bilayer structures we considered them as an additional to the exchange interaction between F and AF layers contribution to the temperature dependence of magnetization dynamics. As far as we know that was not made before.
Another new result obtained through our investigation is the change of the sign of rotatable anisotropy and unusual resonance field behaviour observed when IrMn was changed to FeMn in Co/AF structure.
We also revised the introduction to clarify the goal of the work and novelty of the results (lines 73-107).
Point 2: Can authors provide insight into how the results change with varying thicknesses of FM and AFM layers?
Response 2: Thanks for the question. Variation of the magnetic properties with the thickness of F and AF layers in F/AF bilayers is, of course, a very important issue. It leads to the simultaneous change of many properties of F/AF structures such as exchange bias, rotatable anisotropy, and FMR linewidth together. That is why the point of our paper was to study the effects of different factors on magnetization dynamics at fixed thicknesses of F, AF, buffer and protective layers.
The F and AF layer thickness dependence of the exchange bias has been investigated in 90th and reviewed in classical paper of J. Nogués and I.K. Schuller [1]. In some modern studies (for example [2]) the enhancement of exchange bias at a few nm thick AF layers is observed. FMR linewidth and rotatable anisotropy dependence on the F and AF layers thickness have been studied in a number of papers too, see for example [3, 4]. There non monotonous dependence of FMR linewidth and rotatable anisotropy on AF layer thickness was observed. As for F layer thickness decrease it is shown to increase FMR linewidth in accordance with TMS mechanism.
The effects of layers thicknesses especially in a few nm region on temperature dependence of dynamics in F/AF systems as well as more detailed structural characterization are very valuable and, certainly, could be an issue of another study.
Referenses:
[1] J. Nogués, I.K. Schuller Exchange bias. J. Magn. Magn. Mat., 1999, vol. 192, no. 2: p. 203-232.
[2] Goyat, E.; Behera, N.; Barwal, V.; Siwach, R.; Goyat, G.; Gupta, N.K.; Pandey, L.; Kumar, N.; Hait, S.; Chaudhary, S. Large exchange bias and spin pumping in ultrathin IrMn/Co system for spintronic device applications. Appl. Surf. Sci. 2022, 588 152914.
[3] Rodríguez-Suárez, R.L.; Vilela-Leáo, L.H.; Bueno, T.; Oliveira, A.B.; de Almeida, J.R.L.; Landeros, P.; Rezende, S.M.; Azevedo, A. Critical thickness investigation of magnetic properties in exchange-coupled bilayers. Phys. Rev. B 2011, 83(22), 224418.
[4] Rezende, S.M.; Azevedo, A.; Lucena, M.A.; de Aguiar, F.M. Anomalous spin-wave damping in exchange-biased films. Phys. Rev. B 2001, 63(21), 214418.
Point 3: Fig. 5a shows the opposite behaviour for Ir and Fe case at a temperature less than 200K for the resonance field. What is the reason behind this? Both, these are AFM layers.
Response 3: Thank you for the question. Indeed, the exchange coupling in IrMn and FeMn samples is drastically different. The origin of this difference is discussed in lines 319-321. In short, the exchange coupling at F/AF interface can be either ferromagnetic, producing normal exchange bias and normal rotatable anisotropy for small AF-grains and a left shift FMR line, or antiferromagnetic, producing a negative (right shifted) rotatable anisotropy. We attribute the negative rotatable anisotropy along with unusual resonance field temperature dependence for Co/FeMn sample to the presence of antiferromagnetically coupled grains. Currently, we cannot define unambiguously the origin of that. We can say only a possible origin due to F/AF interface structure, composition, roughness etc. mentioned in the text.

Reviewer 2 Report
In this manuscript, the authors systematically study the temperature dependence of various anisotropy field of the FM/AFM structures. The mechanism of evolution of exchange bias and rotational anisotropy from AFM layers are investigated and proposed to be contributed from nonuniform grain size and weak layer textures. While the paper is quite clear, it could be further improved by the following suggestions and discussions.
1) In the “materials and methods” part, film stacks are described to be coated with native SiO2, I wonder if Si substrate were oxidized before Ta seed layer deposition or oxidized after full film deposition. Also I wonder what the stage temperature during the deposition and if any annealing is applied after growth since samples Hex is relatively low for these thick AFM layers.
In the FMR testing part, a word typo "obtained" in sentence "Temperature dependences of FMR fields for free Co, Co/FeMn and Co/IrMn obtained by in-plane measurements at different EA orientations are shown in Figure 5 ", and same paragraph 5th row, another typo "shows".
2) Authors should present XRD result in a large theta angle scale since in the later part, authors stated that "the texture of the F layer is practically absent in our sample ". For example, I wonder if there is any Cu (200), Cu (220) phases were detected above 50 degrees.
3) From the summarized Table 2, Hk of Co/IrMn is 88Oe from VSM. I wonder what the method you used to estimate the Hk of your second sample Co/IrMn (MH loop comparison or Law-of-Approach-to-Saturation?). If you use simple method like saturation field difference to estimate, Hk of sample 3 (Co/FeMn) could be also estimated by this method, although 100Oe cannot fully saturate the sample and didn't present hard axis loop in Figure 3, but normally VSM testing field should not be limited to 100Oe and could provide higher field to saturate sample.
4) FMR testing are utilized in this manuscript to analyze the performance and properties of samples. However few questions related to this part need to be solved. Firstly, authors should describe more details for this temperature dependent FMR measurement, for example what kind of setup of FMR used here for measurement, coaxial, microstrip line? And what frequency of microwave and bias field were applied during the measurement of all samples, are they all consistent?
It would be better to correlate Meff, damping values from your sample based on the fitting values of the kittel equation (H vs f) rather than use a reference value. Also Ms is preferred to collect from deposited sample, temperature dependency MH loop measurement could be achieved if your VSM setup has that feature especially your lowest temperature is still >100K.
In the FMR testing part, a word typo "obtained" in sentence "Temperature dependences of FMR fields for free Co, Co/FeMn and Co/IrMn obtained by in-plane measurements at different EA orientations are shown in Figure 5 ", and same paragraph 5th row, another typo "shows".
5) In this manuscript, Authors proposed that the temperature dependent HRA and HEB were mainly contributed from the non uniform grain size distribution of AFM films. I wonder what the critical grain size of antiferromagnetic layer is to be considered as stable and ordered grains at certain temperature. It would be preferred to reference some theory or calculation results here to enhance the discussion and analysis. And it would be better to provide the grain size distribution of prepared samples, the presented AFM figures are hard to track the grain size value and distribution of these polycrystalline sample.
And a related concern should be addressed in the discussion part of broaden FMR linewidth. Delta E_FeMn and Delta E_IrMn were estimated by the grain size, however, as paper mentioned, grain size of FeMn is not uniform, I wonder how to properly estimate the grain volume of samples and calulate the Delta E_FeMn.
1) In the FMR testing part, a word typo "obtained" in sentence "Temperature dependences of FMR fields for free Co, Co/FeMn and Co/IrMn obtained by in-plane measurements at different EA orientations are shown in Figure 5 ", and same paragraph 5th row, another typo "shows".
2) In the FMR testing part, a word typo "obtained" in sentence "Temperature dependences of FMR fields for free Co, Co/FeMn and Co/IrMn obtained by in-plane measurements at different EA orientations are shown in Figure 5 ", and same paragraph 5th row, another typo "shows".
Author Response
We thank the reviewer for high evaluation of the importance of our manuscript and helpful suggestions.
We have improved the manuscript text in accordance with the comments.
Point 1: In the “materials and methods” part, film stacks are described to be coated with native SiO2, I wonder if Si substrate were oxidized before Ta seed layer deposition or oxidized after full film deposition. Also I wonder what the stage temperature during the deposition and if any annealing is applied after growth since samples Hex is relatively low for these thick AFM layers.
In the FMR testing part, a word typo "obtained" in sentence "Temperature dependences of FMR fields for free Co, Co/FeMn and Co/IrMn obtained by in-plane measurements at different EA orientations are shown in Figure 5 ", and same paragraph 5th row, another typo "shows".
Response 1: The (100)Si-plates with native oxide were used as the substrates. The ambient temperature was kept during the layers deposition and no thermal annealing was applied after the deposition. The relatively low exchange bias was due to small grains and low fraction of (111) texture in AF layers.
The typos are corrected.
Point 2: Authors should present XRD result in a large theta angle scale since in the later part, authors stated that "the texture of the F layer is practically absent in our sample ". For example, I wonder if there is any Cu (200), Cu (220) phases were detected above 50 degrees.
Response 2: We are sorry, in submitted paper draft a paragraph concerning our XRD data along with grain size evaluation from XRD data was accidently missed. Now we added that (lines 141-155). As mentioned there (lines 142 and 150) there are no another cobalt peaks in thetha-2theta scans. In GI scans no any Co peaks were observed (and certainly there were no Cu lines).
In the attached file, XRD scans in a larger range (>50o) are shown. As can be seen from them, there are no Co peaks at theta > 50o in theta-2theta scans. In GI scans there are some more peaks at theta > 50o, but these peaks are also present in the Si/Ta sample, where there is no cobalt, meaning that they are definitely not from Co. In the paper we show scans in a narrower scan range for better visibility of the peaks related to the F/AF structure.
Point 3. From the summarized Table 2, Hk of Co/IrMn is 88Oe from VSM. I wonder what the method you used to estimate the Hk of your second sample Co/IrMn (MH loop comparison or Law-of-Approach-to-Saturation?). If you use simple method like saturation field difference to estimate, Hk of sample 3 (Co/FeMn) could be also estimated by this method, although 100Oe cannot fully saturate the sample and didn't present hard axis loop in Figure 3, but normally VSM testing field should not be limited to 100Oe and could provide higher field to saturate sample.
Response 3: Thank you for the question. As mentioned in text the HK was evaluated by a simple M-H loops comparison as shown in Figure 4 (a). The VSM scans were done in a larger H-range than 100 Oe, but the M-H plots in a wider H-range do not bring any additional information. Plots in a reasonably narrow range allow to show most essential details of the M-H dependence.
Point 4. FMR testing are utilized in this manuscript to analyze the performance and properties of samples. However few questions related to this part need to be solved. Firstly, authors should describe more details for this temperature dependent FMR measurement, for example what kind of setup of FMR used here for measurement, coaxial, microstrip line? And what frequency of microwave and bias field were applied during the measurement of all samples, are they all consistent?
It would be better to correlate Meff, damping values from your sample based on the fitting values of the kittel equation (H vs f) rather than use a reference value. Also Ms is preferred to collect from deposited sample, temperature dependency MH loop measurement could be achieved if your VSM setup has that feature especially your lowest temperature is still >100K.
In the FMR testing part, a word typo "obtained" in sentence "Temperature dependences of FMR fields for free Co, Co/FeMn and Co/IrMn obtained by in-plane measurements at different EA orientations are shown in Figure 5 ", and same paragraph 5th row, another typo "shows".
Response 4: Thank you for the questions. We extended the description of FMR measurements. We added: “FMR measurements were performed at Brucker ELEXSYS e500 spectrometer. The sample was placed into cryostat and the temperature changed by blowing with liquid nitrogen vapor to provide the cooling in the range from the room temperature down to 115 K. The cryostat with the sample was set in the resonator supplied with the fixed pumping frequency f = omega/2pi = 9,4 GHz. The resonator was placed between the electromagnet poles provided the slowly varied magnetic field aligned along the sample surface plane and scanned in range up to 1500 Oe to reach the resonance condition.” (lines 112-119)
Using the fit with the Kittel equation would be helpful for us but in this case another type of FMR setup is required that allows the frequency instead of field scans. This would be also interesting to study the temperature dependence of MH curves and Ms, however, there was no cooling stage in the VSM available in our study.
The typos are corrected.
Point 5. In this manuscript, Authors proposed that the temperature dependent HRA and HEB were mainly contributed from the non uniform grain size distribution of AFM films. I wonder what the critical grain size of antiferromagnetic layer is to be considered as stable and ordered grains at certain temperature. It would be preferred to reference some theory or calculation results here to enhance the discussion and analysis. And it would be better to provide the grain size distribution of prepared samples, the presented AFM figures are hard to track the grain size value and distribution of these polycrystalline sample.
And a related concern should be addressed in the discussion part of broaden FMR linewidth. Delta E_FeMn and Delta E_IrMn were estimated by the grain size, however, as paper mentioned, grain size of FeMn is not uniform, I wonder how to properly estimate the grain volume of samples and calculate the Delta E_FeMn.
Response 5: Thank you for the suggestions. Currently, we cannot claim for quantitative origin of HRA, HEB dependence on AF grainsize. However, we obtained a clear evidence of an important influence of GS on the exchange coupling, i.e. the smaller grain size in IrMn gives larger HRA. The calculations of critical grain sizes for thermally disordered and unstable grains is an interesting and difficult task and is a topic for current studies in our group.
AFM investigations were used in our work as an additional to XRD qualitative method for grain size evaluation. Now we obtained grain size distributions from AFM images and added them in the paper (Figure 3).
In our calculations of Delta E for Co/IrMn and Co/FeMn structures we used the grain size calculated from XRD results as mentioned in text (line 351).
We also corrected typos in text.

Reviewer 3 Report
The authors of the paper investigate the exchange bias effect at the interface of F/AF layers and its dependence on various factors such as temperature, magnetic field, and layer thickness. They also explore the role of surface morphology and structural properties on the observed magnetization behavior.
Here is some suggestions:
1. Current introduction can be misleading to general readers. It should be better structured and focused on specific research related to current work. For example, considering move the detailed discussion for reference 28-41 to discussion?
2. There are typos in labeling of Figure 1b, should be Co/IrMn instead of Co/FeMn
3. Why the alpha-Ta (110) and beta-Ta(002) in XRD spectra of Figure 1c has different intensity for Co/FeMn and Co/IrMn? Does this means the Ta buffer layer or FeMn has different thickness? If so, how authors can exclude the inflence of thickness to RA and EB ( see a reference for thickness dependent RA: Wang B, Wu P, Bagués Salguero N, et al. Stimulated nucleation of skyrmions in a centrosymmetric magnet[J]. ACS nano, 2021, 15(8): 13495-13503.).
4. The authors cliam: "Figure 2 confirm a larger grain size in Co layer compared with IrMn layer. The FeMn film`s surface also demonstrates the presence of grains larger than in IrMn. Low density of large (clearly visible on AFM images) grains in FeMn layer .... " However, it might not very clear for general readers. It will help if authors can point out these grains and stastically shown the grain sizes.
Overall, the English quality of this excerpt appears good but it is advisable to have the full document proofread as some typo found.
Author Response
We thank the reviewer for high evaluation of the importance of our manuscript.
We have improved the manuscript text in accordance with the comments. The corrections are highlighted in green.
Point 1: Current introduction can be misleading to general readers. It should be better structured and focused on specific research related to current work. For example, considering move the detailed discussion for reference 28-41 to discussion?
Response 1: Thank you for your suggestion. We modified the introduction in accordance with it (lines 73-107).
Point 2: There are typos in labeling of Figure 1b, should be Co/IrMn instead of Co/FeMn
Response 2: Thank you for your comment. We fixed it.
Point 3: Why the alpha-Ta (110) and beta-Ta(002) in XRD spectra of Figure 1c has different intensity for Co/FeMn and Co/IrMn? Does this means the Ta buffer layer or FeMn has different thickness? If so, how authors can exclude the inflence of thickness to RA and EB ( see a reference for thickness dependent RA: Wang B, Wu P, Bagués Salguero N, et al. Stimulated nucleation of skyrmions in a centrosymmetric magnet[J]. ACS nano, 2021, 15(8): 13495-13503.).
Response 3: Thank you for the question. As reported in Sec. 2, the thicknesses of buffer and protective Ta-layers were 30 nm, the same in all the samples. In the Co/FeMn and Co/IrMn samples, we see in XRD both the bottom and top layers of tantalum. The structure of the top layer can be affected by the structure of the layer below it. As can be seen from our results, FeMn and IrMn have a different texture and intensity of this texture, which leads to a difference in the structure of protective tantalum layers and different peak intensities. We added some explanation following this remark in the text (lines 153-155). We also cannot exclude that such a result is also due to a slight difference in the measurement speed.
Point 4: The authors cliam: "Figure 2 confirm a larger grain size in Co layer compared with IrMn layer. The FeMn film`s surface also demonstrates the presence of grains larger than in IrMn. Low density of large (clearly visible on AFM images) grains in FeMn layer .... " However, it might not very clear for general readers. It will help if authors can point out these grains and stastically shown the grain sizes.
Reply 4: We are sorry, in submitted paper draft a paragraph concerning our XRD data along with grain size evaluation from XRD data was accidently missed. Now we added that (lines 141-155). As for AFM it was used as an additional qualitative method for grain size evaluation. We circled the examples of grains in Figure 2 (d) and added grain size distributions along with mean grain size values (Figure 3).
We also corrected typos in the text.

Round 2
Reviewer 2 Report
I think the authors have addressed all my concerns and in particular, the authors have added the discussion of grain size distribution of samples in the revised manuscript. I recommend accept the manuscript in present from and publish in Magnetochemistry.